# TFAM’s Contributions to mtDNA Replication and OXPHOS Biogenesis Are Genetically Separable

**DOI:** 10.3390/cells11233754

**Published:** 2022-11-24

**Authors:** Natalya Kozhukhar, Mikhail F. Alexeyev

**Affiliations:** Department of Physiology and Cell Biology, University of South Alabama, Mobile, AL 36688, USA

**Keywords:** GeneSwap approach, mtDNA replication, mtDNA transcription, mtDNA instability, TFAM

## Abstract

The ability of animal orthologs of human mitochondrial transcription factor A (hTFAM) to support the replication of human mitochondrial DNA (hmtDNA) does not follow a simple pattern of phylogenetic closeness or sequence similarity. In particular, TFAM from chickens (*Gallus gallus*, chTFAM), unlike TFAM from the “living fossil” fish coelacanth (*Latimeria chalumnae*), cannot support hmtDNA replication. Here, we implemented the recently developed GeneSwap approach for reverse genetic analysis of chTFAM to obtain insights into this apparent contradiction. By implementing limited “humanization” of chTFAM focused either on amino acid residues that make DNA contacts, or the ones with significant variances in side chains, we isolated two variants, Ch13 and Ch22. The former has a low mtDNA copy number (mtCN) but robust respiration. The converse is true of Ch22. Ch13 and Ch22 complement each other’s deficiencies. Opposite directionalities of changes in mtCN and respiration were also observed in cells expressing frog TFAM. This led us to conclude that TFAM’s contributions to mtDNA replication and respiratory chain biogenesis are genetically separable. We also present evidence that TFAM residues that make DNA contacts play the leading role in mtDNA replication. Finally, we present evidence for a novel mode of regulation of the respiratory chain biogenesis by regulating the supply of rRNA subunits.

## 1. Introduction

In metazoans, mitochondrial DNA (mtDNA) encodes protein subunits of the oxidative phosphorylation system (OXPHOS), as well as the rRNA and tRNA needed to translate these subunits using a mitochondria-specific genetic code. Therefore, mtDNA in these organisms is central to cellular bioenergetics, which in turn modulates all vital processes.

Mitochondrial transcription factor A (TFAM) is the key player in mtDNA metabolism. It is a member of the HMGB subfamily of high mobility group (HMG)/DNA-binding proteins involved in various functions, including DNA repair, immune responses, and wound healing [1]. TFAM consists of five distinct domains: a cleavable matrix targeting sequence (MTS), which is removed during protein import into the mitochondrial matrix, two HMG homology domains connected by a linker, and a tail (Figure 1). Human and murine TFAM also contain short leader sequences, seven and six amino acids (aa), respectively, located between MTS and HMG1. By inducing sharp (~180°) bends, TFAM mediates mtDNA compaction and assembly into nucleoids [2,3], structures where mtDNA replication and transcription are thought to occur. mtDNA bending by TFAM also was shown to be important to fully activate transcription at the LSP promoter in vitro [4].

Whole-body TFAM knockout (KO) is embryonically lethal and is accompanied by severe depletion of mtDNA and mitochondrial transcripts [5], which indicates the critical contributions of TFAM to mtDNA replication and transcription. In some tissues, mtDNA content is directly proportional to TFAM expression [5]. However, accumulating evidence suggests that this proportionality is not a universal phenomenon and, in an extreme case, mtDNA content can be inversely proportional to TFAM expression [6,7].

According to current models, TFAM, in complex with mitochondrial RNA polymerase (POLRMT) and mitochondrial transcription factor B2 (TFB2M), contributes to OXPHOS biogenesis by transcribing genes encoding OXPHOS subunits as well as RNA components of the mitochondrial translation apparatus [8]. TFAM’s contribution to mtDNA replication is less well understood. However, the consensus is that a fraction of transcripts from the mitochondrial light strand promoter (LSP, identical in sequence to mtDNA heavy strand) are prematurely terminated at the guanine-rich conserved sequence block II (CSBII) to generate primers for mitochondrial heavy strand replication [9,10,11]. This model intimately links TFAM’s contributions to mtDNA replication and OXPHOS biogenesis.

Evidence of the intimate link between TFAM contributions to mtCN control and respiratory chain biogenesis notwithstanding, it has been demonstrated that overexpression of the mutant [8] or wild-type TFAM [9] in mice can elevate mtCN without the corresponding increase in respiration. However, the exact mechanism remains enigmatic.

Our understanding of mtDNA replication in human cells remains limited, in part due to the lack of tractable in vitro or in situ models to study this process. Recently, we developed the GeneSwap reverse genetic analysis method to study mtDNA replication in situ and made a striking observation that TFAMs from the ‘living fossil’ fish coelacanth (*Latimeria chalumnae*) and a frog (*Xenopus laevis*) can support mtDNA replication in human cells, whereas TFAM from phylogenetically less distant chicken (*Gallus gallus*) cannot [10]. This apparent paradox could not be explained by the extent of primary amino acid sequence similarity between proteins because chicken TFAM (chTFAM) is more similar to the human ortholog than TFAMs from either coelacanths or frogs. However, chimeric TFAM composed of either hTFAM N-terminal domain (NTD) and chTFAM C-terminal domain (CTD) or chTFAM NTD and hTFAM CTD supports hmtDNA replication [10]. This observation suggests that the inability of chTFAM to support hmtDNA replication is a cumulative effect of 100 aa residues that are at variance between mature forms of human and chicken proteins (Figure 2) and that it cannot be attributed to any of these substitutions individually. Therefore, each aa substitution in chTFAM is conditionally permissive (the condition here being the context of other substitutions in the same domain). Moreover, neither 67 aa that are at variance in the NTD nor 33 aa that are at variance in the CTD, collectively, can account for the chTFAM deficiency in hmtDNA replication [10].

These unexpected observations motivated the current study, in which we attempted to impart to chTFAM, through mutagenesis, the ability to support the replication of human mtDNA (hmtDNA).

## 2. Materials and Methods

### 2.1. Cell Growth and Treatment

All cells were propagated in Dulbecco’s modified Eagle’s medium (DMEM) containing 4.5 g/L glucose, 10% fetal bovine serum, 50 µg/mL gentamycin, 50 µg/mL uridine, and 1 mM sodium pyruvate in a humidified atmosphere containing 5% CO_2_ at 37 °C. This medium is permissive for the growth of cells devoid of mtDNA (ρ^0^ cells; +UP medium). When indicated, uridine and pyruvate were omitted from this medium for selection against ρ^0^ cells (-UP medium). All cell lines were checked monthly for mycoplasma contamination and semiannually authenticated by STR profiling (Labcorp, Burlington, NC, USA).

### 2.2. DNA Constructs

Plasmids were constructed according to standard recombinant DNA protocols [11]. Synthetic chTFAM variants (Appendix A) were generated with either native or human MTS by Twist Bioscience (South San Francisco, CA, USA). chTFAM MTS is functional in human cells [10]. Synthetic chTFAMs were cloned into pMA4659 (Addgene, Watertown, MA cat. no. 184854, Appendix A). Cre recombinase for the excision of wthTFAM in 143B cells was delivered with the help of rv.3442 (Appendix A, Addgene no. 184852). PhiC31 recombinase for excision of chTFAM variants was delivered with rv.5136, which also encodes hygromycin resistance (Appendix A, Addgene no. 184853). Cells expressing Ch13 or Ch22 variants were complemented with rv.5724, rv.5748, or rv.5749, which encode hygromycin resistance and wt hTFAM, Ch13, or Ch22, respectively (Appendix A). In this manuscript, retroviral vector plasmids begin with the prefix “pMA”, whereas corresponding retroviral particles generated using this vector are designated with the prefix “rv.” and the corresponding number. For example, pMA4659 refers to a retroviral vector plasmid, whereas rv.4659 refers to retroviral particles.

### 2.3. GeneSwap

The general outline of the GeneSwap approach is presented in Figure 1B,C. To implement GeneSwap, 143B#6 cells were co-transduced with rv.3442 (Addgene no. 184852, which encodes puromycin resistance and Cre recombinase), and a retrovirus carrying a chTFAM variant. This resulted in the simultaneous excision of hTFAM and the introduction of a chTFAM variant. The 143B#6 cells were co-transduced in 35-mm dishes at 30% confluence by applying 1 mL of fresh medium and 0.5 mL of each retroviral supernatant overnight. The next day, the medium was removed and replaced with the fresh virus-free medium. Twenty-four hours later, the cells were trypsinized, and 10-fold serial dilutions were plated into 150-mm dishes containing selective medium (+UP medium supplemented with 3 µg/mL puromycin (Millipore Sigma, Burlington, MA, USA) and 1 mg/mL G418 (GoldBio, St. Louis, MO, USA) and grown until colonies appeared. The colonies were picked into 24-well plates, expanded, and analyzed using a DirectPCR reagent (Genprice, Inc., San Jose, CA, USA, cat no. 388-302-C) for the retention of mtDNA, excision of wt hTFAM and introduction of chTFAM variants using primers listed in Appendix A according to schemes presented in Figure 3A–C. Subsequently, rho+ (containing mtDNA) GeneSwap cotransductants were further validated by excising chTFAM variants with PhiC31 recombinase encoded by the retrovirus rv.5136 (Appendix A). A loss of mtDNA as a result of this excision was interpreted in support of the notion that the excised chTFAM variant supports mtDNA replication in human cells.

### 2.4. Retrovirus Generation

Supernatants containing recombinant retroviruses were generated by transiently transfecting packaging Phoenix Ampho cells with genomic plasmids using polyethyleneimine (PEI) with an average molecular weight of 25,000 at a 1:3 DNA:PEI ratio. The supernatants were collected 48 h after transfection, filtered through 0.45 μm filters (Millipore Sigma cat no. SLHAR33SS), and stored at −80 °C until use.

### 2.5. mtDNA Diagnostics

The presence of mtDNA was established by duplex PCR with two pairs of primers, one specific to nDNA and another specific to mtDNA, as described previously [12]. In Figure 3 and Figure 4, nDNA primers were directed against the single-copy nuclear gene POLRMT. Because of the multicopy nature of mtDNA, it eclipses the amplification of the single-copy nuclear gene. As a result, no nDNA amplification is visible in samples containing mtDNA at near-normal mtCN. In Figure 5, nDNA primers are directed against 18S rDNA, which is present in the nuclear genome in >200 copies. This setup increases the sensitivity of PCR reactions. As a result, amplification of both nDNA and mtDNA is observed in samples in which mtDNA is present. Primers are listed in Appendix A.

### 2.6. mtCN Determination by Direct Droplet Digital PCR (dddPCR)

mtCN was determined as described previously [13]. Briefly, cells were collected by trypsinization, counted, and ~10^6^-cell pellets were generated and frozen at −80 °C. Pellets were resuspended in PBS at ~10,000 cells/μL, 10 μL aliquots were removed and mixed with 90 μL of the solution containing 50 μg proteinase K, 40 μL of H_2_O, and 50 μL of the DirectPCR solution (Genprice, Inc., San Jose, CA, USA cat no. 388-302-C), the mix was incubated at 50 °C for 30 min and then at 95 °C for another 30 min, the solution was adjusted to 500 μL with H_2_O, and 3 μL of the resulting solution was used as the template in 20 μL dddPCR reaction to determine nDNA content using primers and probes listed in Appendix A. For mtDNA quantification, nDNA samples were diluted 500-fold, and 3 μL of the resulting dilution was used in 20 μL ddPCR reactions with primers and probes listed in Appendix A. dddPCR reactions contained 0.9 μM of each forward and reverse primer, 0.25 μM probe, 10 μL of the 2× ddPCR Supermix for Probes (No dUTP), 10 units of EcoRI HF restriction enzyme (New England Biolabs, Beverly, MA, USA, cat no. R3101S), and the balance of water. The cycling parameters were: initial denaturation for 10 min at 95 °C, followed by 40 cycles of 20 s at 94 °C + 1 min at 60 °C, followed by 10 min at 98 °C, followed by the hold at 4 °C. Each sample was measured in 2 technical replicas. To calculate mtCN per cell, the concentration of mtDNA targets was multiplied by the dilution factor and divided by 0.5× concentration of nDNA targets. Each mtDNA template concentration was combined with each nDNA template concentration generating four mtCN values for each sample.

### 2.7. PhiC31-Mediated Excision of Proviral Inserts

TFAM variants delivered with the help of rv.4659 derivatives were excised by transducing cells with rv.5136 (Appendix A), which encodes a codon-optimized PhiC31 recombinase (PhiC31°) and hygromycin resistance. Successful excision was verified by PCR genotyping using the primers listed in Appendix A.

### 2.8. Genotyping of hTFAM Excision in 143B#6 Cells

PCR diagnostics of hTFAM excision in 143B#6 cells were conducted using the primers listed in Appendix A according to the schemes presented in Figure 3A–C.

### 2.9. Quantitation of Mitochondrial Transcripts

Quantitation of mitochondrial transcripts was performed by RT‒qPCR using the primers listed in Appendix A. RNA was isolated using an EZ-10 DNAaway RNA Miniprep Kit (Bio Basic, Amherst, NY, USA, cat no. BS88136) and treated with a gDNA removal kit (Enzo Life Sciences, Farmingdale, NY, USA, cat no. ENZ-KIT136-0050) to reduce mtDNA contamination prior to reverse transcription with a SensiFast cDNA synthesis kit (Bioline USA, Taunton, MA, USA, cat no. BIO-65053), which was supplemented with primers for MT-ND6 RT‒qPCR. In most experiments, three transcripts representative of three mitochondrial promoters were quantitated using a SYBR Fast kit (Roche Holdings AG, Basel, Switzerland, cat no. KK4601): MT-ND6 (for the light strand promoter, LSP), MT-RNR2 (for the heavy strand promoter 1, HSP1), and MT-ND1 (for the heavy strand promoter 2, HSP2). Some experiments also included the quantitation of MT-CO1 transcripts. Transcript abundancies were normalized to those of HPRT.

### 2.10. Cellular Respiration

This was evaluated with an XFe24 extracellular flux analyzer (Seahorse Bioscience, Billerica, MA, USA) as described previously [14]. Real-time ATP assay was utilized. This assay uses values of ATP-linked OCR and ECAR to calculate the fractions of ATP produced through respiration and glycolysis, respectively, in a unit of time (https://www.agilent.com/cs/library/whitepaper/public/whitepaper-quantify-atp-production-rate-cell-analysis-5991-9303en-agilent.pdf, accessed on 15 October 2022).

### 2.11. Western Blotting

This was performed as described previously [14]. The antibodies used were anti-MT-CO1 (Abcam, Cambridge, MA, ab14705), anti-MT-CO2 (Abcam, Cambridge, MA, USA, ab91317), and anti-β-actin (Sigma, A5441).

### 2.12. Amino Acid Alignments

These were derived using the MUSCLE algorithm of the MegAlign Pro program Lasergene 17 package (DNASTAR Inc., Madison, WI, USA).

### 2.13. Statistical Analyses

These were performed using the GraphPad Prism v.9.1.0 (San Diego, CA, USA) software package.

## 3. Results

### 3.1. A Limited Number of aa Substitutions Enable chTFAM to Support hmtDNA Replication

According to the prevailing model, TFAM interacts with both mtDNA and components of the mitochondrial transcription machinery during the synthesis of the primer for mtDNA replication. It may also interact with the mitochondrial replisome during elongation. Therefore, we were interested in identifying the relative contributions of TFAM-mtDNA and TFAM-protein interactions to the inability of chicken TFAM (chTFAM) to replicate hmtDNA. To broadly address these two extremes, we generated two chTFAM variants. One version, Ch13, had human MTS and significant changes in charge, polarity, or size of the amino acid (aa) side chains at 13 positions in chTFAM, altered to conform to human prototypes (F53L, N71K, E106K, K130D, I140K, Q167E, T174Q, F181K, Q191E, Q193E, A215E, K216Q, Q231K). Another version, Ch22, had 22 aa, which contact mtDNA in available crystal structures of hTFAM and are discordant between human and chicken proteins [4], mutated to human prototypes (Appendix A). Remarkably, both versions supported replication of hmtDNA despite having only three aa substitutions (F53L, T174Q, and F181K) in common (Figure 3D).

### 3.2. F53L, T174Q, and F181K Substitutions Do Not Enable chTFAM Functionality in hmtDNA Replication

The observation of three common aa substitutions in otherwise nonoverlapping sets of mutations that enable hmtDNA replication by chTFAM suggests that these aa may be the key drivers of this newly acquired property. To address this possibility, we generated a chTFAM variant with three aa substitutions (F53L, T174Q, and F181K, Ch3). 143B cells expressing Ch3 did not retain hmtDNA, indicating that, by themselves, these three substitutions do not enable replication of hmtDNA (Figure 3E). Conversely, F53L, T174Q, and F181K substitutions were dispensable for the ability of Ch22 to replicate hmtDNA, as a Ch22 variant lacking these substitutions (Ch22-3) supported hmtDNA replication (Figure 3E). Ch22-3’s proficiency in hmtDNA replication indicates that F53L, T174Q, and F181K substitutions are not critical in the context of the other 19 substitutions in aa making DNA contacts in Ch22. On the other hand, cells expressing the Ch13 variant that lacks F53L, T174Q, and F181K substitutions (Ch13-3) did not support the replication of hmtDNA (Figure 3F). Therefore, by themselves, F53L, T174Q, and F181K substitutions do not enable hmtDNA replication by chTFAM. However, they can do so in the context of an additional 10 aa substitutions found in Ch13. Another conclusion is that nonoverlapping sets of substitutions in chTFAM (as in Ch13 vs. Ch22-3) enable this protein’s functionality in hmtDNA replication. In other words, chTFAM’s inability to support hmtDNA replication cannot be attributed to any particular aa substitution but is rather a cumulative effect of multiple substitutions.

### 3.3. Ch13 Chimeras Do Not Support hmtDNA Replication

In a previous study [10], we showed that even though chTFAM is nonfunctional in hmtDNA replication, a chimera between its N-terminal domain (NTD, Figure 1A) and hTFAM C-terminal domain (CTD) as well as a chimera between hTFAM NTD and chTFAM CTD were functional. Therefore, we examined the functionality of chimeras between wt chTFAM and Ch13 (using functional Ch13 as a surrogate for hTFAM). However, a chimera between Ch13 NTD and wt chTFAM CTD (Ch13/WT) with only 5 NTD substitutions did not support hmtDNA replication. Similarly, a chimera consisting of wt chTFAM NTD and Ch13 CTD (ChWT/13) with 8 CTD substitutions also failed to rescue hmtDNA replication (Figure 3F).

### 3.4. Substitutions in TFAM Residues That Make DNA Contacts Are Responsible for chTFAM’s Inability to Rescue hmtDNA Replication

In the two nonoverlapping sets of mutations that rescue chTFAM’s inability to support hmtDNA replication, there is one common motif: both sets contain aa that make DNA contacts (3 aa in Ch13 and 19 aa in Ch22-3). This suggests that it is differences in aa that make DNA contacts that are responsible for chTFAM’s incapacitation in hmtDNA replication. However, in Ch13-3, only 10 aa out of a total of 78 that do not make DNA contacts were reverted to human prototypes. This left open the possibility that reversing some or all of the remaining 68 aa that do not make DNA contacts and are at variance with hTFAM may rescue chTFAM. Such an outcome would render the above suggestion invalid. Therefore, we tested hTFAM, in which 20 aa that make DNA contacts were mutated to chicken prototypes (hTFAM-20). It was nonfunctional (Figure 3G), leading us to conclude that variances in TFAM residues that make DNA contacts are responsible for chTFAM’s inability to rescue hmtDNA replication.

### 3.5. mtDNA Instability Mediated by the Ch8 TFAM Variant

Crystal structures of hTFAM with mtDNA heavy strand promoter (HSP1), light strand promoter (LSP), or nonspecific DNA (NSP) indicate that in these structures, different sets of aa make contact with DNA [4]. To examine the relative importance of “contacting” aa residues within HSP1 (11 aa), LSP (16 aa), or NSP (13 aa) that are at variance between hTFAM and chTFAM (Appendix A), we generated three versions of chTFAM with these aa differences restored to human prototypes (Ch_HSP1_, Ch_LSP_, and Ch_NSP_, respectively). Unexpectedly, all three new chTFAM variants supported replication of hmtDNA (Figure 4A). Since all three variants supported hmtDNA replication, we were interested in whether chTFAM can be rescued with subsets of substitutions shared by Ch_HSP1_ and Ch_LSP_ (8 aa, Ch8) and/or by all three TFAM variants (4 aa, Ch4) (Figure 4B–D). The Ch4 variant was unable to support hmtDNA replication altogether (Figure 4B). In contrast, the Ch8 variant produced a heterogeneous population of ρ^0^ and ρ^+^ (containing mtDNA) clones. Moreover, ρ^+^ clones were heterogeneous in terms of mtCN (Figure 4D). Expansion of the ρ^+^ clones with low mtCN (e.g., Figure 4D, clone no. 13) in +UP media resulted in the loss of mtDNA. These clones did not survive in the -UP medium, which is selective for mtDNA maintenance (results not shown). Therefore, their cultivation led to the eventual loss of mtDNA (ρ^0^ genotype). However, dilution cloning of a low passage clone no. 13 produced ~19% subclones that retained mtDNA at low mtCN (Figure 5A). In clones with relatively high mtCN, mtDNA was more stable (e.g., clone nos. 2 and 20, Figure 4D). However, even in these clones, continuous cultivation in +UP medium reduced mtCN twofold over 4 weeks (Figure 5C). Additionally, ~10% of the clones produced by dilution cloning lacked mtDNA (Figure 5B). These observations indicate instability of mtDNA in cells expressing the Ch8 TFAM variant.

### 3.6. TFAM’s Contributions to mtDNA Replication and Gene Expression Are Genetically Separable

The restoration of the ability to replicate hmtDNA in both Ch13 and Ch22 was remarkable. Therefore, we examined to what extent mtCN and OXPHOS were restored in each mutant. In Ch13, mtDNA copy number (mtCN) was reduced threefold compared to cells expressing hTFAM, yet mtCN was increased to 186% in cells expressing Ch22 (Figure 6A). Perplexingly, despite the severely reduced mtCN, the % ATP produced through OXPHOS was only modestly decreased in cells expressing Ch13 compared to cells expressing hTFAM. In contrast, cells expressing Ch22 used glycolysis to generate almost all of their ATP despite high mtCN (Figure 6B). In cells expressing Ch22, overexpression of Ch13, Ch22, or even hTFAM suppressed mtCN to various extents but remained at least as high as in cells expressing hTFAM alone (Figure 6A). In cells expressing either Ch13 or Ch22 and complemented with hTFAM, mtCN was almost identical (Figure 6A). Conversely, Ch22, but not Ch13, significantly suppressed OXPHOS in cells expressing Ch13, whereas Ch13 and hTFAM, but not Ch22, rescued OXPHOS in cells expressing Ch22 (Figure 6B). Overall, the OXPHOS capacity correlated well with the expression of MT-CO1 and MT-CO2 proteins (Figure 6C) but not with MT-CO1 transcript levels or transcript levels for MT-ND1 or MT-ND6 (Figure 6D). However, both OXPHOS and steady-state protein levels for MT-CO1 and MT-CO2 correlated with MT-RNR2 levels (Figure 6B–D).

### 3.7. hmtDNA Maintenance and OXPHOS Biogenesis Are Separated in Frog TFAM

To determine whether genetic separation of hmtDNA replication and OXPHOS biogenesis can be observed in naturally occurring TFAM variants, we examined mtCN and respiration in human cells expressing frog (*Xenopus leavis*) or zebrafish (*Danio rerio*) TFAMs instead of hTFAM. Indeed, separation was observed in cells that expressed frog TFAM. In those cells, elevated mtCN was accompanied by severely impaired respiration.

In contrast, in cells expressing zebrafish TFAM, mtCN and respiration were reduced to a similar extent (Figure 7A). Just as was the case with Ch13 and Ch22, differences in steady-state levels of mtDNA-encoded mRNA transcripts were not statistically significant. However, MT-RNR2 levels were significantly reduced in cells expressing frog or zebrafish TFAMs as compared to cells expressing hTFAM, which was also reflected in reduced steady-state levels of MT-CO1 and MT-CO2 polypeptides (Figure 7B,C).

## 4. Discussion

Intuitively, the ability of animal orthologs of human proteins to functionally substitute for human prototypes should decrease with evolutionary distance and aa sequence divergence. Therefore, it was perplexing to observe that TFAMs from mammalian opossum (*Monodelphis domestica*) and Tasmanian devil (*Sarcophilus harrisii*) as well as chicken (*Gallus gallus*) were unable to support the replication of hmtDNA, whereas evolutionarily and sequence-wise more divergent TFAMs from the ‘living fossil’ fish coelacanth (*Latimeria chalumnae*) and frog (*Xenopus laevis*) could [10]. This apparent contradiction motivated us to search for residues within TFAM that are critical for hmtDNA replication.

There are 100 aa at variance between mature forms of human and chicken TFAMs (Figure 1C), of which 22 make DNA contacts in at least one of the three available crystal structures of hTFAM complexed with HSP, LSP, or nonspecific DNA [2,4]. Therefore, we first sought to determine the relative contributions to chTFAM’s inability to support hmtDNA replication of substitutions in aa residues that make DNA contacts vs. those that have major alterations in their side chains. Unexpectedly, the chTFAM variant ‘humanized’ for aa that make DNA contact (22 substitutions) and the variant ‘humanized’ for some major alterations in side chains (13 substitutions) both supported hmtDNA replication despite having only 3 aa alterations in common. These three aa alterations, by themselves, were unable to rescue chTFAM’s ability to replicate hmtDNA and were dispensable for hmtDNA replication in the context of 22 ‘DNA contact’ substitutions. However, they were critical in the context of 13 ‘side chain’ substitutions, suggesting a leading role for aa that make DNA contact in rescuing hmtDNA replication by chTFAM. Indeed, the hTFAM variant, in which 20 aa that make DNA contacts were converted to chicken prototypes, lost its ability to support hmtDNA replication. This line of evidence leads to the first major conclusion of this study regarding the critical role played by TFAM residues that make DNA contacts in supporting mtDNA replication.

If aa make DNA contact play the leading role in ensuring the ability of hTFAM orthologs to support hmtDNA replication, then what DNA sequence determinants do they recognize? To obtain more granular information, we converted chTFAM residues corresponding to those that make DNA contacts at HSP1, LSP, or nonspecific DNA to human prototypes. All three resulting chTFAM variants supported hmtDNA replication. This outcome suggests that substitutions shared by these variants may be critical in enabling hmtDNA replication. However, a chTFAM variant with four substitutions shared by Ch_HSP1_, Ch_LSP_, and Ch_NSP_ did not support hmtDNA replication. Moreover, mtDNA was unstable in cells expressing the Ch8 variant with eight substitutions shared by Ch_HSP1_ and Ch_LSP_. One possibility is that additional, not shared by Ch_HSP1_, Ch_LSP_, and Ch_NSP_, substitutions cooperate in each variant to independently induce conformational changes that enable the rescue of hmtDNA replication. Additional studies will be needed to answer this question definitively.

In chicken mtDNA, MT-ND6 and MT-TE genes are transposed downstream of MT-ND5 as compared to hmtDNA (Figure 2A). As a result, the left border of the control region is formed by MT-TE as opposed to MT-TP. It is unclear whether this rearrangement contributes to the inability of chTFAM to support the replication of hmtDNA. Additional studies are needed to address this question.

In a population of yeast grown to a stationary phase on a fermentable carbon source, approximately 1% of cells would form petite colonies, in which mtDNA is either absent (ρ^0^) or severely deleted/rearranged (ρ^−^) [15]. To our knowledge, similar mtDNA instability has not been described in mammalian cells. In our previous study, we reported modest mtDNA instability in human cells expressing coelacanth (*Latimeria chalumnae*) TFAM [10]. Here, we demonstrated a dramatic mtDNA instability on one of the clones expressing the Ch8 TFAM variant. Collectively, these observations underscore the role played by TFAM in stable mtDNA inheritance.

The prevailing model posits that TFAM contributes to OXPHOS and mtDNA replication by facilitating the transcription of mtDNA-encoded genes and primers for mtDNA replication, respectively. In this model, both contributions are transcriptional, which makes them apparently interdependent so that prematurely terminated LSP transcripts prime mtDNA heavy strand replication. Indeed, the available experimental evidence suggests that mtCN and mitochondrial transcription may go hand-in-hand [5,16]. Therefore, it was unexpected to discover an inverse relationship between changes in mtCN and OXPHOS in cells expressing Ch22 and frog TFAM as well as the different extents to which these parameters were affected in Ch13. These observations clearly indicate that TFAM’s contributions to mtCN and OXPHOS are genetically separable. The rescue of Ch13 deficiency in mtDNA replication by Ch22 and the rescue of Ch22 deficiency in OXPHOS biogenesis by Ch13, as well as an increase in mtCN vs. reduced contribution of respiration to ATP production in cells expressing frog TFAM speak to the same effect.

mtDNA transcription profiles and patterns of MT-CO1 and MT-CO2 polypeptides expression in Ch13, Ch22, frog, or zebrafish TFAMs suggest a model for regulating the expression of mtDNA-encoded polypeptides. According to this model, the differential effects of TFAM mutations on transcription from the HSP1 promoter, which supplies mitochondrial rRNAs, vs. HSP2 and LSP promoters, may be the main driver of the differences in the effects of these mutations on mtDNA replication and OXPHOS biogenesis. A limited availability of HSP1-supplied rRNAs may result in a bottleneck, which would limit the translation rate and steady-state levels of mtDNA-encoded proteins even though their transcript levels may be statistically indistinguishable. Therefore, regulating mitochondrial rRNA supply, e.g., through TFAM posttranslational modifications, may be used for global regulation of OXPHOS biogenesis. Future studies would have to assess the validity of this model.

## Figures and Tables

**Figure 1 cells-11-03754-f001:**
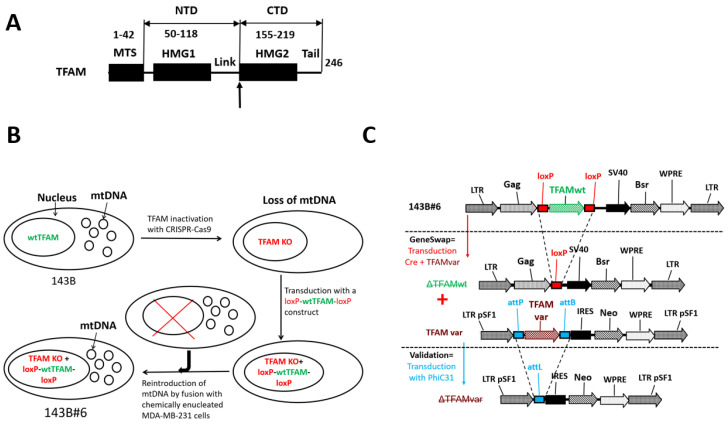
The domain organization of the human TFAM and GeneSwap approach. (**A**) The domain structure of human TFAM. The bold upward vertical arrow designates a crossover point in chimeras (between residues 154 and 155 of the hTFAM). Here, we refer to the TFAM portion preceding the crossover point as the N-terminal domain (NTD) and the following portion C-terminal domain (CTD). Subdomain boundaries are given in aa coordinates. MTS, matrix targeting sequence. (**B**) Derivation of the hTFAM GeneSwap cell line 143B#6. In 143B osteosarcoma cells, TFAM was inactivated with CRISPR/Cas9 and replaced with a retrovirus-encoded, loxP-site-flanked wt TFAM cDNA. To reintroduce mtDNA, the resulting cells were fused with chemically enucleated MDA-MB-231 cells, resulting in the 143B#6 cell line (9). (**C**) Schematics of the GeneSwap approach. To implement GeneSwap, 143B#6 cells (top panel) were co-transduced with retroviruses encoding Cre recombinase and altered TFAM (TFAMvar, central panel). This results in the excision of the wtTFAM cDNA and simultaneous re-expression of TFAMvar. In the resulting cotransductants, mtDNA is retained only if TFAMvar is functional in hmtDNA replication. In that case, the functionality is further validated by transducing cells with PhiC31 recombinase, which affects the loss of TFAMvar and hmtDNA (the bottom panel).

**Figure 2 cells-11-03754-f002:**
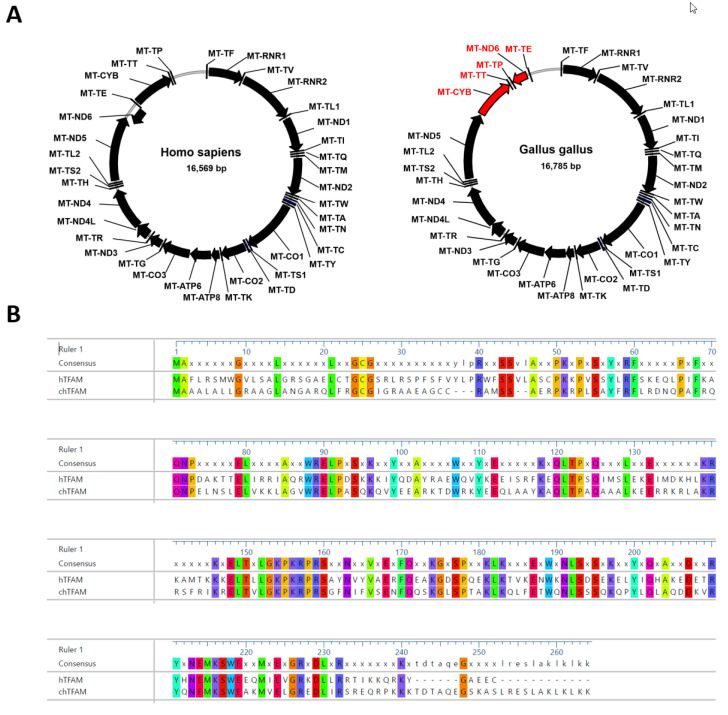
Maps of human and chicken mitochondrial genomes and alignment of corresponding TFAMs. (**A**) Organization of human and chicken mtDNA. Positions of 37 mtDNA-encoded genes are indicated. Note the gene order rearrangement at the 5’ end of the control region in chicken mtDNA (red). (**B**) Alignment of human and chicken TFAMs. Identical residues are highlighted. MegalignPro, Lasergene 17 package, MUSCLE algorithm.

**Figure 3 cells-11-03754-f003:**
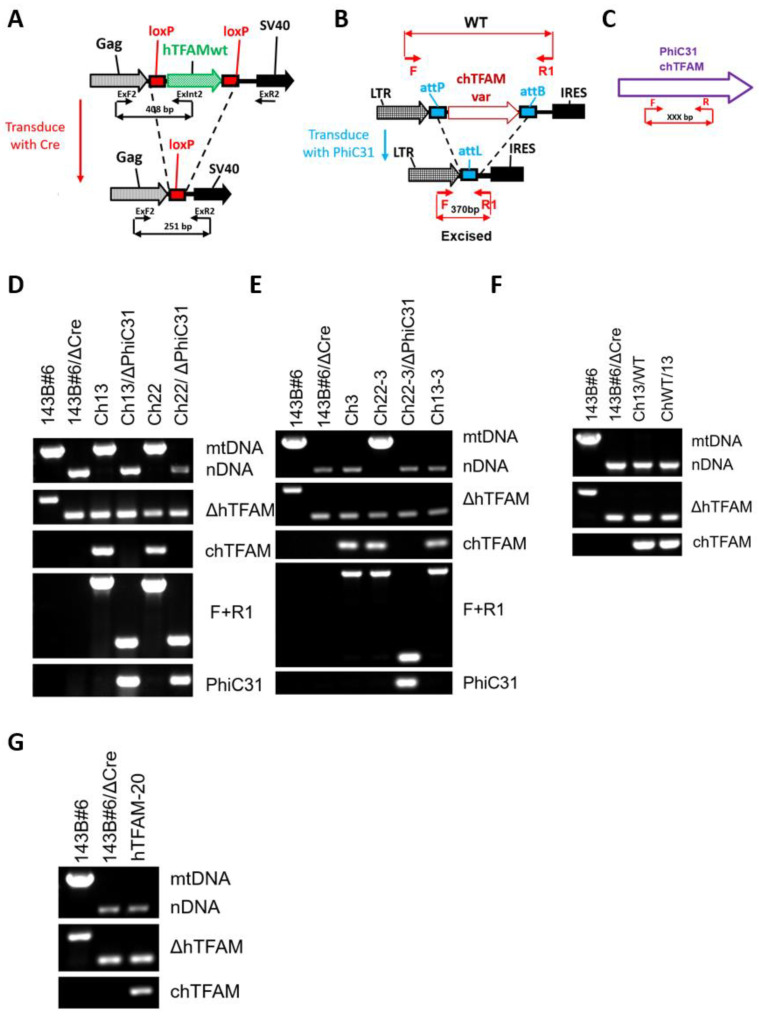
TFAM residues making DNA contacts are critical for mtDNA replication. (**A–C**) PCR genotyping strategies. (**A**) PCR genotyping strategies for excision of hTFAM (ΔhTFAM). (**B**) Genotyping of excision of modified chTFAM (chTFAMvar, F + R1 panel). (**C**) Genotyping of PhiC31 and chTFAM (PhiC31 and chTFAM panels, respectively). mtDNA, nDNA, duplex PCR with two pairs of primers for the detection of hmtDNA; (**D**) Both Ch13 and Ch22 TFAM variants support the replication of hmtDNA. (**E**) Ch22-3 variant supports hmtDNA replication, whereas Ch3 and Ch13-3 variants do not. (**F**) Neither Ch13/WT nor ChWT/13 support the replication of hmtDNA. (**G**) hTFAM with “chicken” DNA contact residues does not support the replication of the hmtDNA.

**Figure 4 cells-11-03754-f004:**
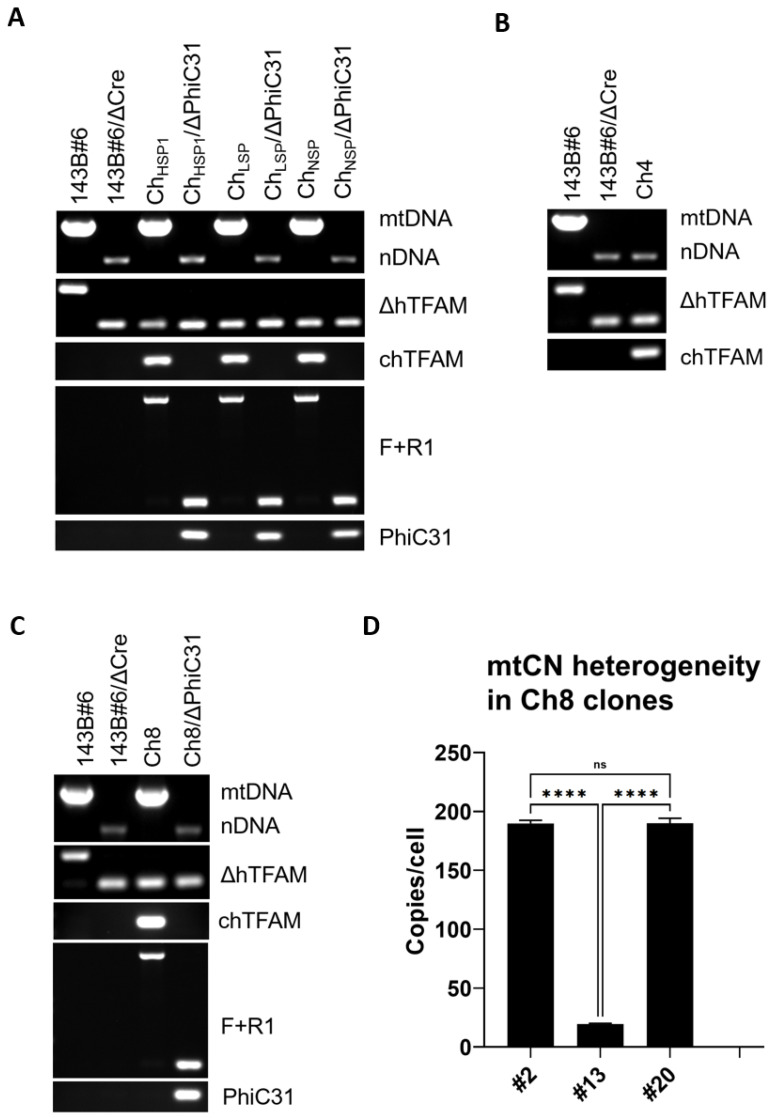
hmtDNA replication by chTFAM variants with substitutions in various groups of residues making DNA contacts. (**A**) Ch_HSP_, Ch_LSP,_ and Ch_NSP_ support hmtDNA replication. (**B**) Ch4 does not support the replication of hmtDNA. (**C**) Ch8 supports replication of hmtDNA. Genotyping strategies are as per Figure 3A–C. Primers are as per Appendix A. (**D**) mtCN heterogeneity mediated by Ch8. Note two clones (no. 2 and no. 20) with relatively high mtCN and one clone (no. 13) with low mtCN. **** *p* < 0.0001, ns, not significant. One-way ANOVA with Tukey post hoc test.

**Figure 5 cells-11-03754-f005:**
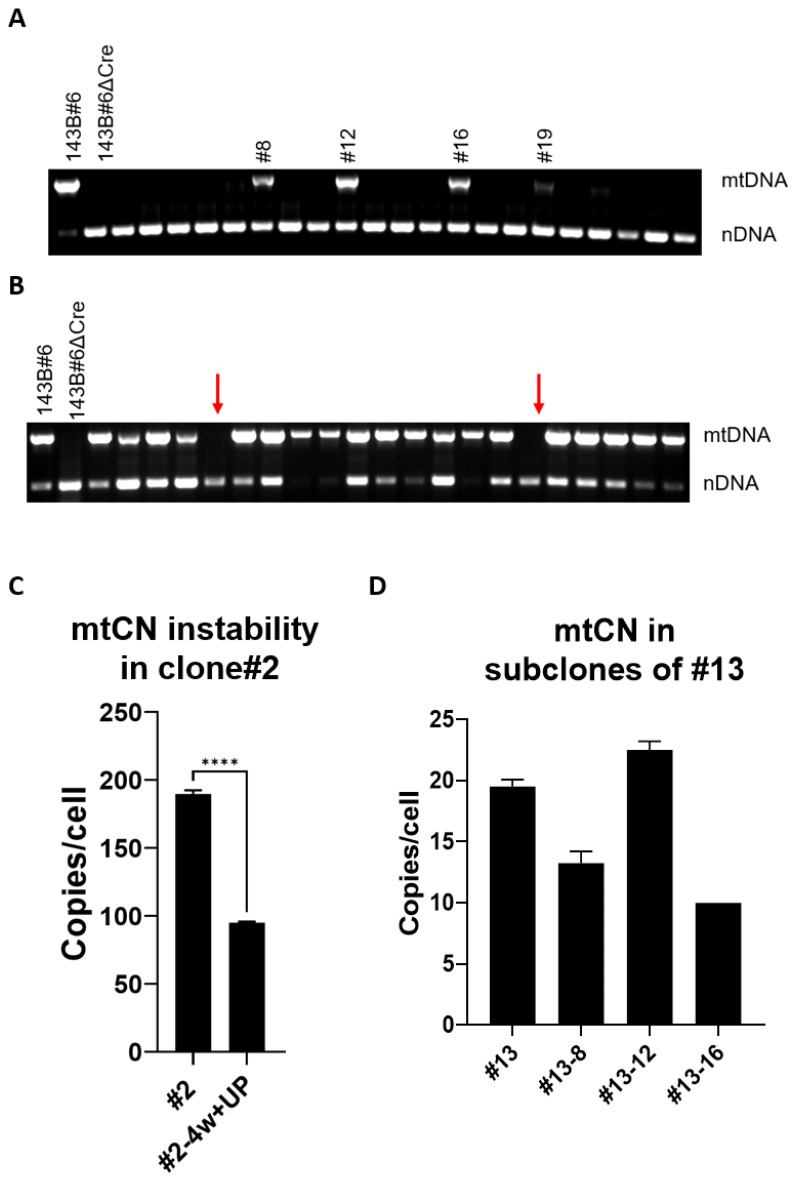
mtDNA instability mediated by the Ch8 TFAM variant. (**A**) Dilution cloning of clone no. 13 with a low mtCN. Approximately 19% of the resulting subclones retain mtDNA. (**B**) Dilution cloning of clone no. 20 with a relatively high mtCN. Note that mtDNA is lost in ~10% of the clones. (**C**) mtDNA instability in clone no. 2. Cells were cultivated for 4 weeks in +UP media, which is nonselective for mtDNA maintenance. Note the twofold reduction in mtCN. **** *p* < 0.0001, two-tailed T-test assuming unequal variance. (**D**) mtCN in subclone nos. 8, 12, and 16 of clone no. 13 (Panel A) is equal to or lower than that in clone no. 13.

**Figure 6 cells-11-03754-f006:**
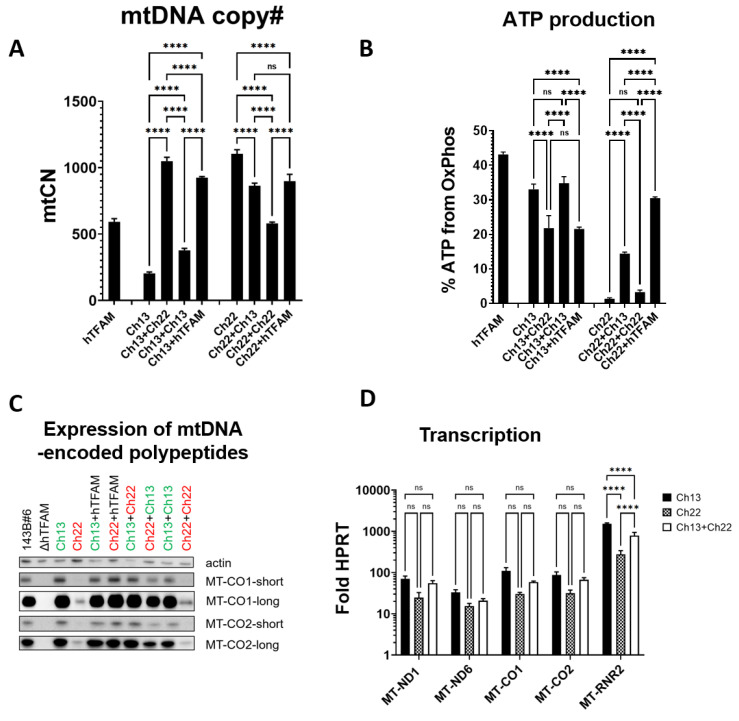
Genetic separation of TFAM’s contributions to mtDNA replication and gene expression. (**A**) mtCN in Ch13 and Ch22 and its rescue with Ch13, Ch22 and hTFAM. ns, not significant; **** *p* < 0.0001. One-way ANOVA with post hoc Tukey test. (**B**) Respiratory activity as indicated by % of cellular ATP generated by OXPHOS. Note the inverse relationship between mtCN and OXPHOS ATP in Ch13 and Ch22 and the patterns of their rescue with Ch13, Ch22, and hTFAM. **** *p* < 0.0001, one-way ANOVA with post hoc Tukey’s test. Representative results of two biological experiments with at least three technical replicas per experiment. (**C**) Expression of mtDNA-encoded MT-CO1 and MT-CO2 in Ch13 and Ch22 and their rescue with Ch13, Ch22, and hTFAM. Short and long refer to exposure times. (**D**) Mitochondrial transcription in cells expressing Ch13, Ch22, and Ch13 + Ch22. Cumulative results of three biological experiments with at least three technical replicates for each transcript in each cell line. ns, not significant; **** *p* < 0.0001. Two-way ANOVA.

**Figure 7 cells-11-03754-f007:**
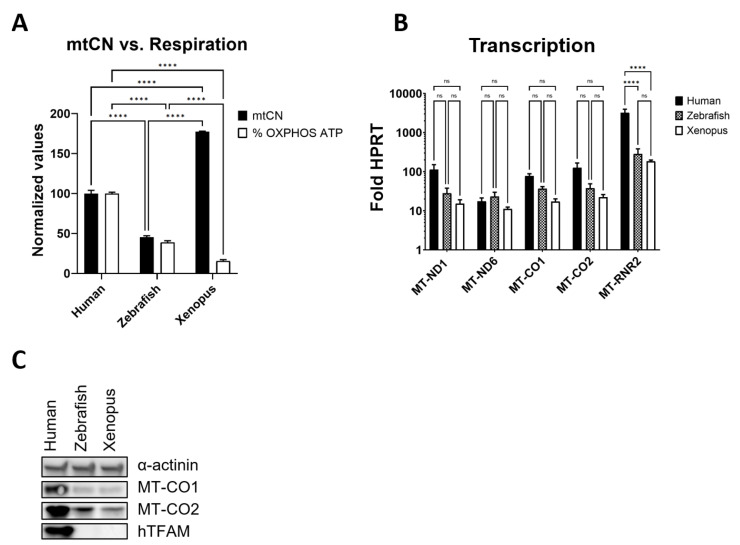
Separation of mtDNA replication and OXPHOS in cells expressing frog TFAM. (**A**) mtCN and respiration in cells expressing TFAM from *Homo sapiens*, *Xenopus leavis*, or zebrafish. Values have been normalized to those in cells expressing human TFAM, which were assigned 100%. Results are representative of at least two biological replicas with at least three technical replicas per biological replica. **** *p* < 0.0001. Two-way ANOVA with Tukey’s multiple comparisons test. (**B**) mitochondrial transcription analysis in cells expressing human, frog, or zebrafish TFAM. Results are an average of three biological experiments with at least three technical replicates per experiment. **** *p* < 0.0001. Two-way ANOVA with Tukey’s multiple comparisons test. (**C**) Western blotting analysis of expression of hTFAM and mtDNA-encoded MT-CO1 and MT-CO2.

## Data Availability

The data presented in this study are contained in the article and Appendix A.

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
