# Peer review of "TFAM’s Contributions to mtDNA Replication and OXPHOS Biogenesis Are Genetically Separable"

_cells, 2022, doi:10.3390/cells11233754_

Round 1

Reviewer 1 Report

The article of Kozhukhar N. and Alexeyev M. highlights physiologically important fact that the participation of mitochondrial transcription factor TFAM in mitochondrial DNA replication and biogenesis of OXPHOS proteins are separated genetically. The authors were motivated by the data that TFAMs from species evolutionarily closer to human (chicken, opossum, Tasmanian devil) were not able to support replication of human mtDNA (hmtDNA) whereas far more divergent TFAMs from fossil fish latimeria or frog (Xenopus laevis) could. Recently developed GeneSwap approach allowed to obtain series of mutant chicken TFAMs (chTFAMs) with limited replacements of aminoacid residues (aa) to human types (“humanization” of chTFAM). The most promising were Ch13 form with 13 changes in TFAM’s side chains and Ch22 form where aa that make contacts with DNA were replaced. Both forms supported hmtDNA replication of hmtDNA. At the same time analyses showed that Ch22 TFAM supported high level of mtDNA transcription and number of DNA copies (mtCN) but low respiration while the opposite was true for Ch13. Moreover Ch13 and Ch22 compensated each other’s deficiences which allowed to conclude that TFAM effects on mtDNA replication and translation of respiratory chain proteins are separated.  An original model of regulation of OXPHOS biogenesis is presented by the authors according to which different effects of TFAM mutations in Ch13 and Ch22 are based on TFAM’s effect on transcription from HSP1 promoter which regulates the availability of mitochondrial rRNA. Evidence was also presented that TFAM residues which make DNA contacts play a leading role in mtDNA replication. Besides Ch8 TFAM mutant form demonstrated DNA instability during cell cultivation. Together with previous data concerning DNA instability in cells with latimeria’s TFAM it is the first precedent of obtaining ρ- mammalian cells.

The article demonstrates the highest technical level of experimental base. All conclusions are confirmed by experimental data. To prove the conclusions a limited number of experiments were made also on cells expressing frog and zebrafish TFAMs. The data are statistically validated. A new original hypothesis of the way by which TFAM may regulate OXPHOS biogenesis is presented.  

Author Response

We are grateful to the reviewer for their kind comments

Reviewer 2 Report

This paper analyzes the recovery of human mitochondrial DNA replication in human cells lacking TFAM protein by expressing variants of chicken TFAM protein. Chicken TFAM is not able to support human mtDNA replication.  The authors conclude that TFAM's contributions to mtDNA replication and respiratory chain biogenesis are genetically separable.

Although the work uses an interesting approach, and the results could be of interest in the mitochondrial field, the study has important limitations and would benefit from a major revision.

It has not been considered that TFAM overexpression can also lower mtDNA copy number, as described “in vivo” in a recent job. (“High levels of TFAM repress mammalian mitochondrial DNA transcription in vivo” Life Sci Alliance. 2021 Aug 30;4(11):e202101034.  doi: 10.26508/lsa.202101034.  Print 2021 Nov.)

It would be interesting to check if all the cell lines generated express the different TFAM variants at similar levels (for example including an epitope for detection). 

Major concerns:

1. Introduction:

The section does not address the important issues to focus the work and contain some vagueness.  

-It is established that mtDNA transcription generates an RNA primer required for initiation of mtDNA replication, however the authors do not include the relevant bibliography.  The possible model described in reference [8] although interesting, is not confirmed in more actual works. 

It should be clarify that TFAM DNA bending is important for promoter function (in association with transcription). Line 39: “By inducing sharp (~180o) 39 bends, TFAM mediates mtDNA compaction and assembly into nucleoids [2,3], structures 40 where mtDNA replication and transcription are thought to occur”.  

-The work does not identify the references in which TFAM interaction with DNA was described. It should clarify the positions of TFAM that interact with DNA in transcription promoters and those that interact non-specifically with DNA. The selection of variations is not well reasoned. In results section reference [10], a molecular cloning laboratory manual is refernced instead of the relevant papers. 

-In my opinion, a previous publication that reaches a similar conclusion: “It is thus possible to experimentally dissociate mtDNA copy number regulation from mtDNA expression and mitochondrial biogenesis in mammals in vivo” should be referenced. (“Mitochondrial transcription factor A regulates mtDNA copy number in mammals”. Ekstrand MI, Falkenberg M, Rantanen A, Park CB, Gaspari M, Hultenby K, Rustin P, Gustafsson CM, Larsson NG. Hum Mol Genet. 2004 May 1;13(9):935-44. doi: 10.1093/hmg/ddh109. Epub 2004 Mar 11.)

-The introduction section includes a reference to Figure 2, it is unusual to have figures cited in the introduction and would be better off in supplementary material. Moreover, Figure 1 is cited later in the text (material and methods section)!. 

-Figure 1 is redundant. It can be found in a previously published work by the authors. 

2. Material and methods:

The procedure used to analyze the production of ATP (Figura 6 B) has not been indicated in the section. 

Line 119: delivered with the help of rv.3442 (Figure S1, Addgene#184852). PhiC31 recombinase for 119.  It should be noted that the rv. refers to retroviral particles.

Line 125: Substitute Figure 1 with refence to the published paper. 

3. Results:

Section 3.1:  In this section the rationale of the selection of variants is not adequately described, the reference [10] is not relevant. The supplementary figure with the variations is not referenced.

Section 3.2: In my opinion section 3.3 should be after section 3.1.  The section 3.2 placed here could be is misleading hear.

Section 3.5: Reference [10] is not adequate for TFAM crystal structures.

Section 3.6:  The result with frog TFAM is interesting and it would be interesting to explore it further. 

Figure legends: In general figure legends should be improved (to be self-explanatory).  

Figure 3: In Panels C, D, E, F, in my opinion the duplex PCR with two pair of primers should give a nuclear amplification in all cases if PCR reagents are not limiting (as in Figure 5 A and B). This is a major concern. It could affect the mtCN measurement.

Figures in panels A and B should be bigger.

Figure 6: In Panel D the numbers in the y-axis of the graph are incorrect. “ns” should be indicated in the figure legend instead of repeated in the figure.  Mitochondrial rRNAs are usually at least 10 times more abundant than mitochondrial mature mRNAs and are easier to measure. Could that account for the significant differences? Could the very low values for mRNAs abundance affect statistical significance?

4. Discusion:

A hypothesis about which chicken or frog amino acid changes (related to human) may affect separation of functions is missing. How could they alter the transcription but not generation of primers for replication?

Reviewer 3 Report

Kozhukar et al, employed an interesting genetic approach called geneswap technique to understand whether role of TFAM in mtDNA replication and OXPHOS biogenesis separable or not. they also used limited humanization of chTFAM and found indeed the mtDNA replication and oxphos biogenesis could be a separable function of TFAM. overall the study sound and the authors approach seem very interesting and novel. however, authors carried our very little experiments mostly PCR based analysis to make a big claim. some of the claims should be strengthened by doing additional experiments and also adding better controls. 

1. First of all, presence of chTFAM variants inside mitochondria should be shown by doing mitochondrial isolation followed by mitoplast preparation. 

2. Authors should perform in vitro mtDNA replication assay using chimera TFAM variants claimed to have different effect on hmtDNA replication.

3.  mitochondrial copy number protocol is missing in materials and methods. are they normalized to nuclear coded gene? 

4. does expression of TFAM variants like ch13 and ch22 has any effect on stability or expression of other mtDNA replicative proteins?

5. does expression of TFAM variants like ch13 and ch22 affect mitochondrial parameters like OCR or ECAR?

6.  if the authors claim differences between the clones and chimeras, they should do statistical analysis to show significance in difference (all  the mtCN's)

7. figure 7, graph size is different than others

Round 2

Reviewer 2 Report

I would like to clarify that in my version of the original manuscript, ref #10 (in line 467) was referenced on lines 191 and 261. That motivated my Criticisms #3, #5, #12 and #14.

Line 467:  10. Sambrook, J.; Russel, D.W. Molecular Cloning. A laboratory manual; Cold Spring Harbor Laboratory Press: New York, 2001. 467

Section 3.1.- 

Line 190: discordant between human and chicken proteins [10], mutated to human prototypes 191 

Section 3.5.- 

Line 261: of aa make contact with DNA [10]. To examine the relative importance of "contacting" aa 262

Criticisms #16 

My concern about the duplex PCR and the mtCN measurement is resolved now. It seems appropriate to me to reduce the extended dddPCR protocol on the revised manuscript to the reference (DOI: 10.1016/j.mito.2021.09.014) (as in the original manuscript).

Reviewer 3 Report

Authors provided sufficient explanation for the comments. Manuscript can be accepted in present form